# Biological Assessment of the NO-Dependent Endothelial Function

**DOI:** 10.3390/molecules27227921

**Published:** 2022-11-16

**Authors:** Hasnae Boughaleb, Irina Lobysheva, Flavia Dei Zotti, Jean-Luc Balligand, Virginie Montiel

**Affiliations:** 1Pole of Pharmacology and Therapeutics (FATH), Institute of Experimental and Clinical Research (IREC), Université Catholique de Louvain (UCLouvain), 1200 Brussels, Belgium; 2Département de Médecine Interne, Cliniques Universitaires Saint-Luc, 1200 Brussels, Belgium

**Keywords:** nitric oxide, NO-dependent endothelial function, NO bioavailability, endothelial dysfunction, nitrosylated hemoglobin, HbNO, reactive oxygen species, electron paramagnetic resonance spectroscopy, cardiovascular diseases, endothelial nitric oxide synthase

## Abstract

Nitric oxide (NO) is implicated in numerous physiological processes, including vascular homeostasis. Reduced NO bioavailability is a hallmark of endothelial dysfunction, a prequel to many cardiovascular diseases. Biomarkers of an early NO-dependent endothelial dysfunction obtained from routine venous blood sampling would be of great interest but are currently lacking. The direct measurement of circulating NO remains a challenge due by its high reactivity and short half-life. The current techniques measure stable products from the NO signaling pathway or metabolic end products of NO that do not accurately represent its bioavailability and, therefore, endothelial function per se. In this review, we will concentrate on an original technique of low temperature electron paramagnetic resonance spectroscopy capable to directly measure the 5-α-coordinated heme nitrosyl-hemoglobin in the T (tense) state (5-α-nitrosyl-hemoglobin or HbNO) obtained from fresh venous human erythrocytes. In humans, HbNO reflects the bioavailability of NO formed in the vasculature from vascular endothelial NOS or exogenous NO donors with minor contribution from erythrocyte NOS. The HbNO signal is directly correlated with the vascular endothelial function and inversely correlated with vascular oxidative stress. Pilot studies support the validity of HbNO measurements both for the detection of endothelial dysfunction in asymptomatic subjects and for the monitoring of such dysfunction in patients with known cardiovascular disease. The impact of therapies or the severity of diseases such as COVID-19 infection involving the endothelium could also be monitored and their incumbent risk of complications better predicted through serial measurements of HbNO.

## 1. Introduction

The vascular endothelium is one of the largest organs in the body and consists of an active cell monolayer producing a wide range of vascular homeostatic mediators. Nitric Oxide (NO) is the main endothelium-derived substance that maintains the vasculature in a quiescent state by inhibition of contractile tone, cellular proliferation, and thrombosis.

NO was firstly discovered by Joseph Priestley in 1772 [1] as a gas denominated “nitrous air”, but it was not until 1979 that Gruetter highlighted the vasodilator function of NO by demonstrating that precontracted bovine coronary arteries relaxed after exposure to a gas mixture of NO [2]. Few years later, Furchgott found that the ability of rabbit thoracic aorta to relax after exposure to acetylcholine was dependent on the presence of an intact vessel intima, concluding that relaxation involved endothelial cells [3] releasing an endothelium-derived relaxation factor (EDRF). Two independents studies [4,5] in 1987 finally identified NO as this EDRF involved in vascular smooth muscle relaxation through the activation of soluble guanylyl cyclase (sGC). The significance of NO-cGMP was recognized by the 1998 Nobel Prize in Physiology and Medicine awarded to Professors Furchgott, Ignarro and Murad for their discoveries on NO as a signaling molecule in the cardiovascular system [6].

Today, the most commonly accepted functional feature of endothelial dysfunction refers to abnormalities in the regulation of the vessel lumen with impaired NO bioavailability [7]. Such endothelial dysfunction is the initiating step towards the development of atherosclerosis, a major cause of mortality and morbidity accounting for up to 31% of deaths worldwide in 2015 [8]. Impairment of the endothelial function is therefore recognized as the prequel to any cardiovascular and metabolic disease associated with hypertension, atherosclerosis, diabetes, hypercholesterolemia [9].

Since NO plays a major role in the maintenance of endothelial function, early detection of its reduced bioavailability would be of great interest to prevent cardiovascular events. Nevertheless, an easy quantification of NO in circulating blood is precluded by its very short half-life (approximately 2 milliseconds) related to its chemical reactivity, itself influenced by the redox status in the vasculature. A reliable assay of the bioavailability of NO in human circulation in vivo would be of high interest both for the screening of high-risk, yet asymptomatic cardiovascular patients and for the monitoring of existing endothelial dysfunction. Such an assessment would also be of interest as a surrogate biomarker to quantitatively measure the effects of new drugs developed to treat endothelial dysfunction.

In this review, we will concentrate on biological methods that have been developed so far to measure the bioavailability of NO (or derivatives thereof) in the human vasculature. We will particularly focus on a quantitative original technique of low-temperature electron paramagnetic resonance (EPR) spectroscopy to measure the 5-α-coordinated heme nitrosyl-hemoglobin in the T (tense) state (commonly named HbNO) that was recently correlated with the development of endothelial dysfunction before any clinical manifestation of cardiovascular disease and proposed as a marker of this impairment.

## 2. Vascular Sources of NO

Circulating NO is mainly regulated through the activity of endothelial nitric oxide synthase (eNOS) but also from the nitrite-dependent NO synthesis, while nitroso/nitrosyl species in biological compounds can also act as a reservoir in the bloodstream (Figure 1).

### 2.1. Oxygen-Dependent Nitric Oxide Synthesis

In the vascular system, NO is mainly produced from the conversion of L-Arginine to L-Citrulline by the constitutive vascular endothelial NOS isoform (eNOS). However, all three NOS isoforms are potentially expressed in various cell types composing the vascular tissue and act in a coordinated manner through their appropriate cellular–subcellular localization. In blood vessels, eNOS is mainly localized in the caveolae of endothelial cells, while the neuronal NOS (nNOS) is expressed in the sarcoplasmic reticulum from vascular smooth muscle cells (VSMC); in addition, the inducible NOS (iNOS) is expressed in the cytosol of all cell types upon exposure to inflammatory cytokines such as Interleukin 1 beta (IL1-beta) and Interferon-gamma. Both eNOS and nNOS are constitutively expressed but are strongly regulated by transcriptional, post-transcriptional and post-translational mechanisms, including S-glutathionylation, S-nitrosation (which both inhibit its function) or protein–protein interactions, such as the inhibitory interaction of eNOS with caveolin-1 at the plasma membrane [10].

Both these enzymes constantly produce small amounts of NO in the vessel wall and in the lumen—particularly eNOS, which is activated by shear stress and stretch [11]. In contrast, the inducible iNOS, once expressed, produces much higher amounts of NO that is mainly involved in nitrosative reactions associated with pathological vascular remodeling [12].

The functional eNOS is organized as a homodimer and needs oxygen and cofactors such as reduced nicotinamide-adenine-dinucleotide phosphate (NADPH), Flavin adenine dinucleotide (FAD), Flavin mononucleotide (FMN) and tetrahydro-L-biopterin (BH4) to be catalytically active [13]. Full eNOS activation requires the binding of Ca^2+^, Calmodulin and a number of kinases in order to phosphorylate the enzyme on Ser1177, its main activation site (e.g., by protein kinase A, B, C, G, AMP-activated protein kinase and Calmodulin-Dependent Protein Kinase II) and phosphatases to dephosphorylate the negative regulatory site on Thr495 [14]. Under physiological circumstances, and as long as eNOS remains dimeric, NO signaling participates to the vascular homeostasis through a cyclic guanosine-5-monophosphate (cGMP)-dependent or via a cGMP-independent pathway [15,16].

eNOS is mainly expressed in endothelial cells, but is also found in platelets [17] and erythrocytes [18]. Endothelial eNOS-derived NO appears as the central regulator of vascular homeostasis by playing crucial roles both in the vascular wall by relaxing and inhibiting VSMC hypertrophy, promoting angiogenesis, but also by diffusion into the vessel lumen where it inhibits leukocyte adhesion, platelet aggregation and thrombosis. An additional involvement in vascular relaxation is provided by nNOS through an Angiotensin II- dependent stimulation mainly in hypertension [19].

Through a functional, calcium-dependent, eNOS-like enzyme, erythrocytes also produce, store and distribute NO in peripheral tissues with a potential role in hypoxia-induced vasodilation [20], platelet inhibition [21] and regulation of red blood cells (RBC) deformability and aggregation [22]. While some contradictory results have questioned the specific role of erythrocyte NOS [23], its involvement is supported by decreased erythrocyte-released NO metabolites (nitrite and nitrate) levels in response to NOS inhibition [24]. A specific contribution of erythrocyte eNOS to blood pressure regulation and nitrite homeostasis has gained recognition from experiments using bone marrow transplantation in chimeric mice competent or deficient for eNOS expression in circulating blood cells [25]. Erythrocyte eNOS expression and function have also emerged as independent risk factors of cardiovascular disease [26].

### 2.2. Nitrite-Dependent Nitric Oxide Synthesis

An alternative pathway can contribute additional NO from the inorganic anions, nitrate (NO_3_^−^) and nitrite (NO_2_^−^). Diet, particularly from vegetables, is the main provider of exogenous nitrate [27] and from cured meats (∼30%), of nitrite, to a lesser extent [28]. Nitrate is reduced into nitrite through the nitrate reductase activity of the bacteria in the oral cavity and enters the stomach where it undergoes an acid-dependent reduction into nitrous acid (HNO_2_) [27]. Nitrous acid then decomposes into nitrogen species, which are partially absorbed into the circulation and recycled to form NO or other bioactive nitrogen species. The reduction of nitrite to NO is enhanced by ascorbic acid and polyphenols, both reducing compounds commonly found in vegetables and fruits. After dietary intake, under normoxic conditions, the L-Arginine-NOS pathway is the main supplier of NO oxidation products from the rapid oxidation of NO by heme proteins (e.g., hemoglobin, to form nitrate with production of methemoglobin) and by cytochrome c oxidase and ceruloplasmin (to form nitrite) [29]. In addition, NO auto-oxidation can also occur [30,31]. Nitrite is considered as a reservoir of NO that can be reduced to bioactive NO through the nitrite reductase activity of a number of proteins, particularly under hypoxic and acidic conditions [28], while hemoglobin and myoglobin would act as oxygen sensors. This mechanism may therefore serve as a backup system mainly when oxygen-dependent NOS function is limited [32].

### 2.3. Nitroso/Nitrosyl Species

As a free radical, the NO “nitrosyl” moiety can easily react either with carbon, nitrogen, oxygen or sulfur atoms, resulting in a nitrosation reaction (where the incorporated NO is referred to as a nitroso group) [33]. NO reaction with a metal-centered protein (e.g., the catalytic site of a metalloenzyme) will induce a nitrosylation reaction (where the incorporated NO is referred to as nitrosyl group) [34]. Protein S-nitrosation (R-SNO) with the thiol group of a cysteine residue has emerged as a main posttranslational modification involved in the regulation of cell signaling proteins (including eNOS as reported above) while repeated transnitrosation reactions enable the incorporation of NO groups into other organic molecules [35]. These compounds naturally occur in vivo; stabilize NO; potentiate its biological effects and mostly act as a storage pool for NO (particularly S-nitroso-Albumin, S-nitroso-Hemoglobin and S-nitrosated proteins) [36].

### 2.4. Red Blood Cells

Erythrocytes have been considered as “NO sinks” due to different interactions be-tween NO and hemoglobin depending on its state. Hemoglobin exists in two distinct conformational states with less affinity for oxygen in the T-state (tense) than the R-state (relaxed). The T-state corresponds to the deoxy-form of hemoglobin and is also known as deoxyhemoglobin (deoxy-Hb) [37]. A dioxygenation reaction in which NO reacts with oxygenated hemoglobin (oxy-Hb) induces the formation of inactive methemoglobin and nitrate, which cannot be converted back to nitrite in the circulation. An interaction of NO with deoxy-Hb produces both iron-nitrosyl hemoglobin (NO bound to the heme group; HbNO) and S-nitroso hemoglobin (NO bound to the thiol group of cysteine 93 in the β-globin chain) [38,39]. These two reactions have both high association and slow dissociation rates supporting the theory of a “NO sink” in erythrocytes and hemoglobin as scavenger of NO [38,40]. However, in vivo, pressures gradients push erythrocytes into the center of the blood vessel, creating a cell-free zone that attenuates the interaction between endothelial cells, NO and hemoglobin, thereby limiting scavenging [41]. Moreover, NO consumption by erythrocytes is 1000 times slower than free hemoglobin, suggesting intrinsic factors limiting NO consumption, including extracellular diffusion limitations [42]. In addition, two pathways have been suggested to mediate the export of NO from the erythrocytes via band 3, either in a metabolon complex with deoxyhemoglobin promoting its nitrite reductase activity [43] or by band 3-specific transnitrosation from S-nitroso-hemoglobin to other NO acceptors outside the cell [44]. 

These propositions combined with the production of NO from the erythrocyte NOS and of a significant amount of NO derivatives (including S-nitrosothiols and S-nitroso-hemoglobin) preserving NO from binding to oxyhemoglobin argue against the NO “scavenging” in erythrocytes. 

Thus, erythrocytes should rather be considered as NO transporters reflecting vascular NO contents and, in a limited fashion, as NO generators through their eNOS activity, albeit with marginal influence on erythrocyte NO content (cf. Section 6.2).

## 3. Endothelial Dysfunction and the Vascular Redox Status

A uniform definition of endothelial dysfunction seems difficult given the multiplicity of its physiological functions, including vascular tone, permeability, anticoagulation and angiogenesis, where altered endothelium-derived NO bioavailability plays a central role. An impaired endothelium-dependent vasodilation is an independent predictor of adverse cardiovascular events and long-term outcomes [45], where the elevated oxidative stress and/or inflammation are the main contributors of the dysfunctional eNOS-dependent NO signaling.

Oxidative stress activates pleiotropic signal transduction pathways ultimately impairing vascular endothelial function and acts by causing nonspecific oxidative damage to DNA, proteins, lipids and macromolecules [46]. Enzymatic sources of reactive oxygen species (ROS) within the vasculature mainly include the nicotinamide adenine dinucleotide phosphate (NADPH) oxidase (Nox1, 2, 4 and 5), the uncoupled eNOS and the respiratory chain of mitochondria from endothelial cells and leucocytes; additional oxidant species are produced from the xanthine oxidoreductase and from the cyclooxygenases in the arachidonic acid metabolism [47]. In the endothelium, inducible Nox1, Nox2 and Nox5 (in humans only) are located at the plasma membrane and involved in endothelial dysfunction, inflammation and apoptosis, whereas the constitutively expressed Nox4, mainly located in the perinuclear zone, would participate in vasoprotection by increasing the NO bioavailability [48]. 

Oxidative stress occurs when excess ROS are generated and cannot be adequately countered by antioxidant systems. Thus, imbalance in the redox environment induces either the presence of a NOS inhibitor such as asymmetric dimethylarginine [49,50], a lack of substrate (through induction of Arginase-1) or cofactors (BH4) for eNOS [51,52] and finally an accelerated NO degradation by ROS per se. 

ROS include mainly free radicals, such as superoxide anion (O_2_^−^) and hydroxyl radical (OH^−^), and nonradical, more stable molecules such as hydrogen peroxide (H_2_O_2_), which, in turn, can generate free radicals. O_2_^−^ is very short-lived owing to its rapid reduction to H_2_O_2_ by superoxide dismutase (SOD), while H_2_O_2_ is itself converted by glutathione peroxidase and catalase to water [53]. In addition to its dismutation to H_2_O_2_, O_2_^−^ directly inactivates NO while forming the powerful oxidant peroxynitrite (ONOO^−^) [54]. Peroxynitrite then oxidizes BH4, which uncouples NOS and converts it into a dysfunctional O_2_^−^-generating enzyme that contributes to vascular oxidative stress, thereby creating a vicious circle [55].

However, due to its relative stability, H_2_O_2_ acts as a second messenger and plays a key role in both the maintenance of vascular homeostasis and endothelial damage related to exacerbated redox signaling, depending on its concentration [56]. H_2_O_2_ mainly acts by post-translational protein modification through the oxidation of thiol residues, creating disulfide bonds and mixed disulfides between glutathione and the thiol of another protein (S-glutathionylation) or with amides to form sulphenylamides (-SN-), justifying a permanent thiol redox control mechanism in endothelial cells through the action of the thioredoxin and glutaredoxin systems [57]. Vascular remodeling mainly occurs when altered proteins belong to cell signaling, such as transcription factors or protein kinases, especially the mitogen-activated protein kinases (MAPK) [58]. 

Intriguingly, membrane-bound vascular O_2_^−^ producing enzymes (Nox1-2-5 and NOSs) can generate H_2_O_2_ after O_2_^−^ dismutation, while the membrane-bound SOD3 is extracellular [59]. Indeed, Nox-derived O_2_^−^ is produced extracellularly and rapidly dismuted to H_2_O_2_ outside of the cell. While the conventional view postulated that H_2_O_2_ freely diffuses back into cells, such passive diffusion is sluggish at best. Instead, we and others demonstrated that specific isoforms of the water channels, Aquaporins, facilitate the rapid and targeted transport of H_2_O_2_ across the cell membrane. Among such few isoforms of aquaporin, labeled as “peroxiporins”, we showed that AQP1, expressed at the membrane of cardiac myocytes transports H_2_O_2_ through its water pore, thereby enabling the activation of an oxidant-dependent pathway leading to hypertrophy and fibrosis in the myocardium [60]. AQP1, highly expressed in human erythrocytes, also controls the intracellular oxidative stress in the same cells [61]. Since we found a high expression of AQP1 also in endothelial cells [62], it is tempting to speculate a similar role of this channel in the regulation of endothelial oxidant stress and endothelial dysfunction. This would open the possibility to prevent or reverse such dysfunction with specific blockers of AQP1 conductance, with much higher specificity than systemic antioxidants, which all failed in previous clinical trials.

## 4. NO Bioavailability Biomarkers

NO formation itself is difficult to measure quantitatively because of its high reactivity and very short half-life in vivo [63]. However, NO reacts with different molecules to form more stable products/metabolites that can be used as surrogates of NO availability.

### 4.1. The Nitrate/Nitrite

NO_2_^−^/NO_3_^−^ (NOx) are quantifiable using spectrophotometric assays. Moreover, nitrite is positively correlated with endothelial function in healthy volunteers [64]. However, measurement of NOx in whole blood is difficult because of the high reactivity of NO_2_^−^ with hemoglobin, as detailed above. After centrifugation and separation of erythrocytes, the NOx are stable in the plasma fraction. The spectrophotometric assay is based on the specific reaction of NO_2_^−^ with the Griess reagent to form a diazonium ion, which is then coupled with N-(1-naphthyl)-ethylenediamine (NED) to form an azo compound quantifiable by spectrophotometry and other colorimetric assays (fluorometry and chemiluminescence). Nitrate can also be measured by this technique after its reduction to nitrite by a nitrate reductase. 

However, a critical appraisal of the Griess assay also showed its limitations [65]. The results with this assay can be highly influenced by dilution of the samples; the materials of the container and the buffer used in the measurement that is different from plasma; precipitation of azo compounds, which makes the liquid turbid and renders the measurement by spectroscopy difficult; uncertain stability of the bacterial nitrate reductase over the time in some kits increases variability. All these variables in preanalytical steps may compromise the accuracy of NOx measurements. 

Another issue is that NO_3_^−^ quantification can only partially reflect endogenous NO availability considering dietary intake, renal clearance and liver metabolism. Moreover, measurements of total NO_2_^−^/NO_3_^−^ do not accurately reproduce the NO production rate.

### 4.2. The Cyclic Guanosine Monophosphate 

Cyclic guanosine-3′,5′-monophosphate (cGMP) is a ubiquitous intracellular second-messenger generated in response to the production of NO and natriuretic peptides (NPs). The effector of the cGMP-dependent NO pathway is the soluble guanylyl cyclase (GC) that produces cGMP. The effects of cGMP are mainly mediated by cGMP-dependent protein kinase G (PKG) and cGMP-regulated phosphodiesterases (PDEs) [66]. cGMP is therefore used as surrogate for NO synthesis and can be measured directly in plasma by several methods such as radioimmunoassay, enzyme-linked immunoassay (ELISA) and by liquid chromatography [67]. The plasma cGMP concentration is positively correlated with nitrite levels in healthy male volunteers [68]. 

However, all these techniques do not discriminate the NO-independent cGMP formation upon NP activation [69]. Moreover, the NO-generated cGMP catabolism is closely regulated by the PDEs that catalyze the breakdown of cGMP into GMP. Some PDEs such as PDE5A and PDE9A that are particularly upregulated during chronic cardiovascular disease are responsible for an accelerated degradation of cGMP unrelated to the NO formation [70,71,72]. 

Therefore, these catabolic pathways may induce changes in the concentration of cGMP that weaken the relationship between circulating cGMP and NO, leading either to an overestimation or an underestimation of the NO originally produced.

### 4.3. The Nitrosyl-Hemoglobin and Others NO Species

Nitroso/nitrosyl species act as a circulating pool of longer-lived NO metabolites. Some metabolites, such as R-SNO species, could even serve to restore NO bioavailability [73] in cardiovascular disorders associated with endothelial dysfunction. These metabolites can be detected by classical colorimetric methods with poor sensitivity, while gas phase chemiluminescence seems the most reliable assay [74]. 

Beside their main role in NO storage within the vasculature, a depletion of plasma nitros(yl)ated species (RXNOs, including S-nitrosothiols, N-nitrosamines and iron-nitrosyl species) determined by reductive gas-phase chemiluminescence has been associated with an endothelial dysfunction (measured by high-resolution vascular ultrasound) [75]. Although these results were promising, other methods allowing a more refined analysis of the different NO species were needed in order to find specific biological markers of the NO-dependent endothelial dysfunction.

Nitrosyl-hemoglobin (HbNO) belongs to the nitrosyl species, and its specific detection can be performed either indirectly by chemiluminescence or directly by electron paramagnetic resonance spectroscopy.

In the early 2000s, an indirect method of detection of HbNO by chemiluminescence assay was described for the indirect measurement of HbNO [76]. Blood was collected in healthy volunteers after NO inhalation and peripheral i.v., and radial artery catheters were placed to collect blood hourly. After blood centrifugation, RBC were washed and were lysed in nitrite-free molecular biology grade water (control solution) or in KCN and K_3_Fe(CN)_6_. After 30 min of incubation, RBC extracts were passed in desalting column to remove nitrite, small thiols, KCN and K_3_Fe(CN)_6_. Then, hemoglobin-containing samples were drawn in contact with I_3_^−^ reactant to release NO for subsequent chemiluminescence detection. The total value of HbNO was obtained from the I_3_^−^-generated signal difference between samples in control and KCN/K_3_Fe(CN)_6_. However, this technique takes time and needs some manipulations. Moreover, RBC are processed over 30 min, a timeframe corresponding to the recently determined stability of the HbNO complex (cf. Section 5.3).

Electron paramagnetic resonance (EPR) spectroscopy directly measures the transition of an unpaired electron in an applied magnetic field and is used for the quantitative detection of paramagnetic molecules, including NO, in mouse, rat and human venous blood ex vivo. While the detection of NO itself remains a challenge due to its low concentration and short half-life, nitrosyl species formed by interaction with hemoglobin (as HbNO) are more stable and EPR-detectable. 

In 1992, a Japanese group used a spin-trapping agent, CO-hemoglobin (Hb-CO), injected in vivo in the peritoneal cavity of rats and, after the collection of arterial blood, were able to detect HbNO identified as the characteristic triplet hyperfine structure in the EPR spectra [77]. 

Nitrosyl-hemoglobin and, more particularly, the 5-α-coordinated heme nitrosyl-hemoglobin in the T (tense) state (HbNO) do not simply act as a reservoir of NO but quantitatively reflect circulating bioavailable NO. Accordingly, we will next focus on a state-of-the-art EPR technique for the measurement of HbNO.

## 5. Electron Paramagnetic Resonance Technique for Measurement of Fresh Frozen Erythrocytes HbNO

### 5.1. There Are Several Paramagnetic Forms of Nitrosyl-Hemoglobin 

The full EPR spectrum of nitrosyl hemoglobin depends on both the ambient oxygen tension of the hemoglobin and the NO-bound heme subunit (α or β) to ultimately correspond to the overlap of three different spectra comprising the 5-coordinated α-HbNO (T-state), the 6-coordinated α-HbNO (R-state), and the 6-coordinated β-HbNO (R-state) [76]. The β-chain heme nitrosyl is always 6-coordinated and is not affected by hemoglobin conformation while the α chain heme nitrosyl can be 5- or 6-coordinated [78]. In the R (oxygenated) conformation, α-nitrosyl heme is 6-coordinated, with four nitrogen ligands from the heme, one histidine residue and one NO giving a poorly resolved EPR signal. However, transition from the R to the T (deoxygenated) state induces a cleavage of the bond between the iron of the heme and the proximal histidine residue with donation of the electron density of Fe (II) to the NO donor species. This reaction leads to 5-coordinated α-hemoglobin with a specific EPR signal showing a hyperfine triplet structure displayed at the g value 2.0 (A _hyper fine structure_ = 16.8 G) [78,79]. 

The 5-α-coordinated nitrosyl-heme (T-state), commonly named HbNO, was therefore chosen according to its relative stability (20 min) and predominance compared to the R-state in venous blood, but also its typical EPR spectrum with a stable, well-resolved triplet hyperfine structure as a dynamic marker of NO availability in the systemic circulation. A more precise identification of this NO-derivative would instead be defined as 5-α-coordinated heme nitrosyl-hemoglobin in the T (tense) state.

### 5.2. EPR Signal Subtraction Procedure to Unmask the Hyper Fine (hf) Complex

An analysis of HbNO EPR signals from human freshly frozen erythrocytes remains difficult due to overlapping EPR signals with others paramagnetic species (Cu-containing protein such as Ceruloplasmin and, mainly, protein-centered free radicals (PFR) with g-values around 2.005) combined with low physiological levels of HbNO (lower threshold limit of 200 nmol/L) [78]. Higher physiological HbNO signals are detectable in rodent erythrocytes with values around 425 nM [80] in mice and above 1000 nM in rats [81]. In humans, such high nitrosyl hemoglobin concentrations were previously observed after NO inhalation [76] or under pathological inflammatory circumstances [82] but not under physiological circumstances. However, the observation that blood levels of HbNO correlated with eNOS activity in isolated vessels prompted towards an improvement of the EPR technique order to use HbNO as a biomarker of circulating NO [80].

A first method of EPR signal subtraction was developed in rodents by administration of the NOS inhibitor, L-NAME in order to reveal a pure HbNO signal [81]. This procedure was refined for human erythrocytes by using antioxidants and subtracting the PFR-dependent EPR component from the target signal to reveal the hyperfine triplet structure of HbNO (Figure 2). This new subtraction approach allowed the quantification of the unmasked EPR signal of HbNO at values lower than the previous detection limit of 200 nmol/L. Addition of an antioxidant mixture (ascorbate and N-acetyl-cysteine, both at 5 mM) to intact erythrocytes in freshly drawn venous blood improved signal detection and, importantly, did not introduce any bias (e.g., artificial increase) in the measurement of the HbNO signal using this subtraction technique [83].

### 5.3. Optimal Conditions of Fresh Frozen Erythrocytes HbNO Preservation for Diagnostic Use

A study in 2018 examined, in detail, the optimal conditions for blood sampling, preparation and HbNO quantification [23]. Upon the addition of a NO-donor, the formation of HbNO was favored in hypoxic (1% of O_2_) compared to normoxic conditions (21% O_2_). Moreover, in freshly drawn human RBCs with preformed HbNO in vivo, low oxygen tension ex vivo preserved HbNO stability. This is expected from the higher propensity of deoxy-Hb to form the 5-α-coordinated heme nitrosyl compound in the T state.

HbNO is also more stable at 20 °C compared to 37 °C with a lower rate of degradation at 20 °C. Moreover, stability was improved in acidic pH (6.2–4.7) compared to physiological pH (7.4–7.2). Based on these data, for the optimal measurements of HbNO, venous blood should be processed under 30 min at 20 °C, including 10 min of centrifugation at 800 g followed by the immediate freezing of erythrocytes in liquid nitrogen for subsequent low-temperature EPR spectroscopy.

## 6. HbNO in Clinical Research

### 6.1. HbNO Is Correlated with Endothelial Function in Healthy Volunteers

Reactive hyperemia measured by peripheral arterial tonometry (PAT) is an accepted non-invasive technique to assess peripheral microvascular endothelial function. The technique measures changes in digital pulse volume during reactive hyperemia induced by deflation of an occluding forearm cuff [84]. NO is known as the main factor responsible for this reactive hyperemia, since the administration of an eNOS inhibitor significantly attenuates the hyperemic response [85]. The Framingham reactive hyperemia index (FRHI), a widely used reactive hyperemia score, is calculated from the natural logarithm of the ratio of post-deflation to baseline pulse amplitude in the hyperemic finger, divided by the same ratio in the contralateral finger serving as the control. An inverse correlation was observed between cumulated cardiovascular risk factors and the FRHI index, as a surrogate of NO-dependent endothelial function [86]. 

In a cohort of healthy volunteers of both sexes (*n* = 50, mean age: 24.7 ± 0.8 years old), our group found a correlation between the EPR-measured HbNO levels in freshly frozen erythrocytes and endothelial function assayed by FRHI (r = 0.58, *p* < 0.0001), with mean erythrocyte HbNO concentrations at the baseline around 219 ± 12 nmol/L. Moreover, the hyperemic response was associated with dynamic changes of HbNO levels in erythrocytes isolated 1 and 2 min after cuff deflation, resulting in dynamic increases of the erythrocytic HbNO signals to 120 ± 8% of the basal levels [83]. Therefore, HbNO measurements in human venous erythrocytes are correlated with endothelial function measured by tonometry during hyperemia.

### 6.2. HbNO Reflects NO Formation from the Vasculature In Vivo

The respective contributions of eNOS in the vascular wall or in erythrocytes as predominant source of blood HbNO is still a matter of debate and has specifically been examined in few studies so far. A specific role of the erythrocyte-derived NO would be supported by experiments showing that, upon inhibition of erythrocyte Arginase 1 (that competes with eNOS for their common substrate, L-arginine) [87], NO production by erythrocytes perfused ex vivo can modulate heart function during ischemia. Our group approached this question in depth by modulating the enzymatic activity of eNOS from both sources ex vivo and in vivo and sequential measures of HbNO production. Briefly, the study confirmed that under inhibition of Arginase-1, HbNO complexes can be generated from human (and mouse) erythrocytes ex vivo, which are abrogated upon NOS inhibition (with N(G)-nitro-L-arginine methyl ester or L-NAME). However, the erythrocyte HbNO signals preformed in vivo were not influenced by sequential treatment with the NOS inhibitor ex vivo, suggesting minor contribution of the erythrocyte NOS to the overall HbNO content. Conversely, the in vivo administration of L-NAME (in mice) significantly decreased the EPR signal of HbNO underlining the contribution of vascular eNOS activity. The implication of endothelial cells in HbNO production was supported by an increase up to 70% of the HbNO signal in erythrocytes from mice genetically deficient in caveolin-1 (expressed in endothelial cells but not in erythrocytes), a well-established allosteric inhibitor of endothelial eNOS, compared to their wild-type littermate [80]. 

These observations argue in favor of the expression of an erythrocyte eNOS that has a constitutive activity [22,26] but minor influence on the whole HbNO formation in erythrocytes, suggesting that the HbNO complex may reflect the bioavailability of NO formed in the vasculature. In addition, the study showed that the erythrocyte HbNO content is also influenced by intravascular administration of an NO donor in vivo [23].

### 6.3. HbNO Is Influenced by the Vascular Redox Balance

Erythrocytes are exposed and sensitive to exogenous and endogenous sources of ROS that profoundly influence their primary function, i.e., the transport and delivery of oxygen through the circulatory system to peripheral tissues. Membranes Nox1-2 and uncoupled eNOS are the main sources of ROS in human erythrocytes [12] and can cause erythrocyte dysfunction, which can be recovered by NO donor supplementation [88]. To prevent this dysfunction, erythrocytes and endothelial cells are equipped with an extensive intracellular antioxidant defense system, including enzymes such as catalase, glutathione peroxidase and different isoforms of cytosolic peroxiredoxins [89]. 

Our group and others [61,90] demonstrated that O_2_^−^ produced by Nox1-2 decreases the steady-state HbNO signals, while inhibition of these ROS-generating enzymes (including uncoupled eNOS) and the erythrocyte antioxidant arsenal protect HbNO levels from degradation. Peroxiredoxin-2, the most abundant antioxidant enzyme in erythrocytes is mainly involved in endogenous H_2_O_2_ degradation, while extracellular membrane-bound catalase participates in exogenous H_2_O_2_ degradation. Specifically, we showed that H_2_O_2_ generated extracellularly via dismutation of O_2_^−^ produced by the plasmalemma-bound NADPH oxidase critically influences steady-state HbNO levels in erythrocytes [55]. Accordingly, HbNO measurements could be used as indicators of both vascular oxidant stress and biomarkers of early endothelial dysfunction.

### 6.4. HbNO Is a Biomarker of Oxidant Stress and Endothelial Dysfunction under Oral Contraceptive Pill

Large cohort studies established an association between the use of oral contraceptive pills (OCP) and adverse cardiovascular events such as venous thromboembolism and/or arterial thrombosis [91]. These conditions can lead to fatal diseases, such as acute myocardial infarction, pulmonary embolism or stroke [92]. Previous studies assessing flow-mediated dilation of the brachial artery suggested a relationship between OCP consumption and the occurrence of premature endothelial dysfunction and atherosclerosis [93,94]. 

In a previous study, our group measured HbNO levels from freshly frozen erythrocytes and vascular redox status in a cohort of young women (*n* = 114, mean age: 22.3 ± 0.2 years old) consuming OCP that contained ethinyl estradiol and various synthetic progestogens, compared to pill-free controls subjects [95]. OCP consumption induced higher mean blood pressure, total cholesterol and triglyceride levels while EPR spectroscopy revealed significantly lower erythrocyte HbNO levels compared to pill-free women (162 ± 8 and 217 ± 12 nmol/L, respectively; *p* < 0.05). These low HbNO values were inversely correlated with vascular oxidative stress characterized by increased plasma peroxide levels and reduced erythrocyte thiols. Interestingly, digital reactive hyperemia pulse tonometry (assessed by endoPAT/FRHI) was significantly decreased only in users of OCP containing drospirenone, which also showed the lowest HbNO levels (Figure 3).

These results mainly highlight the negative impact of OCP in young women on vascular oxidative stress with an early endothelial dysfunction before any clinical manifestation. An assay of HbNO could therefore be used as a biomarker to identify women with endothelial dysfunction at risk of developing thromboembolic disease and may prompt consideration of a change in contraceptive method.

### 6.5. HbNO Is a Biomarker of Vascular and Respiratory Degradation in COVID-19 Patients

SARS-CoV-2 targets endothelial cells via its membrane-anchored angiotensin II-converting enzyme-type 2, leading to subsequent endotheliosis that could explain, at least in part, the widespread thrombosis and microangiopathy during COVID-19 [96,97]. Endothelial dysfunction as a precursor to subsequent injury was suspected in infected patients with profound alterations of the microcirculation and endothelial glycocalyx, which were correlated with clinical parameters and disease outcome [98]. Previous observations already demonstrated reduced plasma NO species in COVID-19 patients compared to controls [99] while others suggested a monitoring of NOx levels [100] during the infection as well as restoring NO through dietary inorganic nitrate [101]. 

In order to assess the NO-dependent endothelial function under SARS-CoV-2 infection, the HbNO levels and vascular redox status were measured in COVID-19 patients hospitalized in intensive care unit (ICU) and compared with levels in a noninfected control group matched for similar cardiovascular risk factors, as well as in COVID-19 patients hospitalized in a (non-ICU) general ward. An additional comparison was made with patients hospitalized in ICU for septic shock non related to COVID-19 [102]. The results showed a NO-dependent endothelial dysfunction proportional to the severity of COVID-19 with an imbalance between exacerbated oxidative stress (reflected by increased plasma lipid peroxides) and a reduced NO bioavailability defined by low HbNO signal. In a limited number of COVID-19 patients, HbNO levels decreased between their initial admission to the general ward and their later transfer to the ICU, suggesting worsening of endothelial dysfunction that may participate to the increased severity of their COVID-19 disease. Moreover, HbNO levels decreased beyond levels observed in the control group matched for cardiovascular risk factors, suggesting additional aggression from the virus itself. Conversely, patients with septic shock had sharply increased blood HbNO values, a first-in-human observation in this paradigmatic disease with cytokine-dependent induction of iNOS. 

These results identify HbNO as an early biomarker of NO-dependent endothelial dysfunction during COVID-19 that may also improve the detection of COVID-19 patients at higher risk of vascular complications. 

### 6.6. HbNO as Biomarker for Detecting the Development of a Cardiovascular Complications during or after Surgery 

Endothelial dysfunction is a prequel to cardiovascular complications [103,104]. Based on the previous clinical observations reported above, HbNO could be used as a biomarker to assess cardiovascular risk in patients undergoing a surgical operation with general anesthesia that, by itself, entails a cardiovascular stress. A study aimed at validating HbNO as a predictive biomarker of per- and post-operative cardiovascular complications is under way in 1500 patients with different risk profiles evaluated preoperatively. This variety of patients will allow to refine the estimation of normal and critical levels of HbNO according to the patient’s health status and to examine the additional value of HbNO to refine cardiovascular risk independently from traditional risk factors.

### 6.7. Limitations of HbNO

While the technical conditions for sampling, preanalytical preparation, storage and EPR spectra analysis have thoroughly been studied and described, accurate measurements of HbNO in a clinical setting still need careful adherence to a well-specified protocol, including rapid centrifugation of freshly drawn blood and immediate freezing of erythrocytes. Failure to respect such critical steps may induce artefacts or inability to measure the HbNO signal, in part due to inadvertent blood reoxygenation and/or methemoglobin formation. Likewise, routine application of the subtraction protocol for HbNO unmasking and quantification requires professional skills in EPR spectroscopy. Caution is also needed in the interpretation of the obtained measurements, given potential confounding factors, such as systemic inflammation (as illustrated in sepsis) or exogenous/endogenous donors of NO species, such as nitrite, that may contribute part of the HbNO signal, particularly in hypoxic/acidic conditions.

## 7. Conclusions

Despite these caveats, previous studies, as reported above, would support erythrocyte 5-α-coordinated heme nitrosyl-hemoglobin (HbNO) as a valuable biomarker of NO bioavailability and NO-dependent endothelial function. These studies showed an inverse relationship between HbNO and the early preclinical stage of endothelial dysfunction related to vascular oxidative stress, which, in turn, can promote atherosclerosis leading to cardiovascular disease. In humans, the erythrocyte HbNO complex mainly reflects the activity of the vascular eNOS or other NO sources in vivo while contribution of the erythrocyte eNOS seems minor. An original low-temperature EPR spectrum subtraction technique quantifies HbNO in freshly frozen erythrocytes from a simple venous blood sample easily provided in a clinical setting under specific technical conditions. HbNO measurements would allow both the identification of endothelial dysfunction in asymptomatic subjects at high cardiovascular risk and, potentially, better stratification of this risk beyond traditional factors. Longitudinal follow-up with HbNO measurements may also allow to monitor the influence of various therapeutic, life-style and dietary measures for disease prevention. As a sensitive and dynamic marker of NO production in the vasculature, HbNO measurements would also allow to monitor the effect of exogenous NO administration, including in the form of a nitrate-enriched diet (with subsequent reduction to nitrite by enterosalivary microbes) or with any new drugs promoting NO production.

HbNO may provide information on the patient’s tolerance to potentially vasculo-toxic treatments, such as specific anticancer or anti-inflammatory “inibs” and guide medical decisions to interrupt some medications (e.g., oral contraceptive pills). Finally, HbNO may help to refine the prognosis of diseases, such as COVID-19 infection or others whose progression may be driven by NO-dependent endothelial dysfunction (Figure 4).

## Figures and Tables

**Figure 1 molecules-27-07921-f001:**
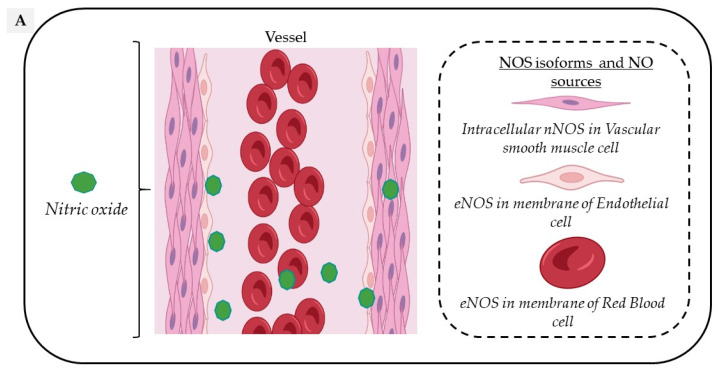
Vascular sources and “reservoirs” of NO. (**A**) NO, produced by eNOS, diffuses to VSMC and the lumen of vessels. (**B**) In the lumen, NO reacts with albumin to form S-nitroso-albumin (1), with OxyHb to form MetHb and nitrate in normoxic conditions (2) and with deoxy-Hb to form HbNO (3) and S-nitroso-Hb in hypoxic conditions (3). S-nitroso-albumin (1) and S-nitroso-Hb (3) serve as storage pools of NO. NO can also be oxidized into nitrite under normoxic condition (4). Combined with the dietary intake, nitrite, in turn, it serves as the storage pool and can be reduced to NO under hypoxic and acidic conditions (4). In VSMC, NO is also produced by nNOS. deoxy-Hb: deoxygenated hemoglobin; EC: endothelial cells, eNOS: endothelial Nitric Oxide synthase; HbNO: nitrosylated hemoglobin; MetHb: methemoglobin; nNOS: neuronal Nitric Oxide synthase; NO: nitric oxide; NO_2_^−^: nitrite; NO_3_^−^: nitrate; OxyHb: oxygenated hemoglobin; RBC: red blood cells; S-nitroso-Hb: S-nitroso-hemoglobin; VSMC: vascular smooth muscular cells.

**Figure 2 molecules-27-07921-f002:**
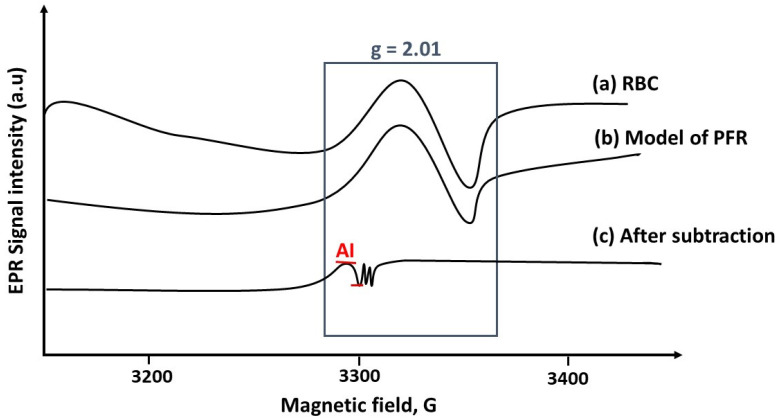
Electron Paramagnetic Resonance (EPR) spectra. The graph represents the whole EPR spectrum of human frozen RBC (**a**), the model spectrum of protein-centered free radicals (PFR) (**b**) and the final EPR spectrum after subtraction of the PFR model (**c**). The peak-to-peak amplitude of hyperfine component AI of the HbNO triplet structure corresponding to 5- coordinated alpha-HbNO that is used for quantification is illustrated in the final spectrum (**c**) [83].

**Figure 3 molecules-27-07921-f003:**
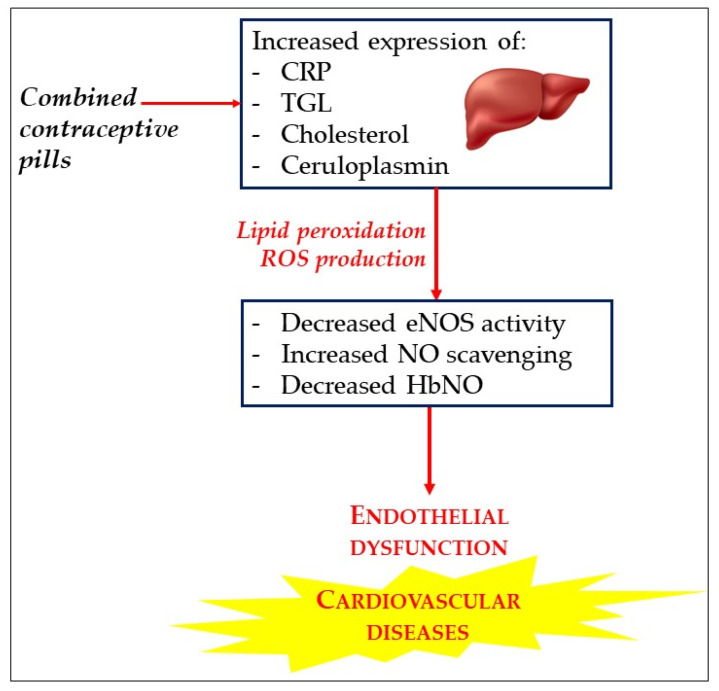
Impact of oral combined contraceptive pill on endothelial function. Women consuming contraceptive pills exhibit an increase in the expression of C-reactive protein (CRP), triglycerides (TGL), cholesterol and ceruloplasmin. This is accompanied with ROS overproduction and an increased lipid peroxidation, resulting in decreased eNOS activity (possibly by oxidation of BH4 and uncoupling of eNOS) and increased NO scavenging. This is reflected by a decreased level of HbNO. A NO-dependent endothelial dysfunction appears that increases the likelihood of developing cardiovascular disease.

**Figure 4 molecules-27-07921-f004:**
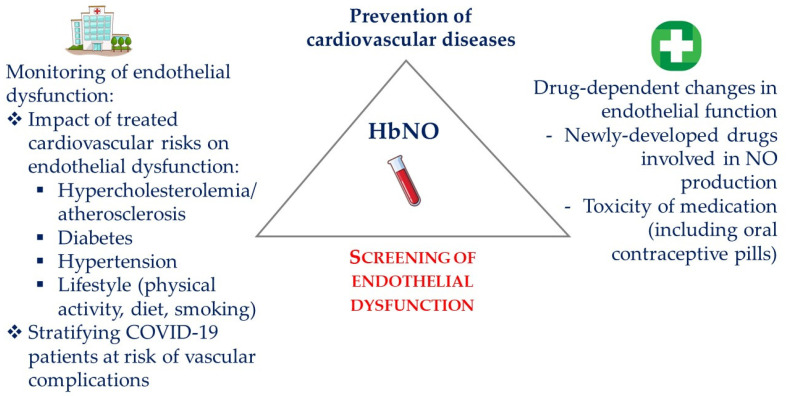
Applications of HbNO measurements in the management of cardiovascular healthcare. Screening of endothelial dysfunction using HbNO measurements in clinical routine may guide the prevention of cardiovascular diseases. Monitoring HbNO levels may also help in evaluating the efficacy of preventive measures towards the reduction of cardiovascular risk factors (thereby improving patient compliance), the efficacy of newly developed drugs that improve NO bioavailability, as well as in monitoring the tolerance to vasotoxic drugs. HbNO may also improve the stratification of disease severity as in COVID-19 and prevent cardiovascular complications.

## Data Availability

Not applicable.

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
