# Peer review of "Biological Assessment of the NO-Dependent Endothelial Function"

_molecules, 2022, doi:10.3390/molecules27227921_

Round 1

Reviewer 1 Report

A well-prepared manuscript devoted to the analysis of modern methods NO control of under endothelial dysfunction is presented. Considering the for application prospects of an presented original technique of low temperature electron paramagnetic resonance spectroscopy capable to directly measure the 5 α -coordinate-heme nitrosyl-hemoglobin in the tense-state from fresh venous human erythrocytes,  I believe that article is useful for a wide range of specialists, and will be interesting to readers.

There is technical note on the manuscript:

Figures 1,3 and 4 are better to give in improved quality (reduce their size and increase the fonts of the inscriptions in them, etc.) for rational use of space.

Author Response

Response: We thank the reviewer for his/her comments. We have changed the quality of the graphics so that they are more informative and the informations mentioned more readable.

Reviewer 2 Report

This is a review regarding the impact of NO on vascular function, with emphasis on HbNO.

Concerns:

1. The figures are of low quality and do not depict concepts presented. For example, figure 1 is difficult to read since the font size and colors chosen by the authors make the fonts unreadable. There is no legend and it is unclear what point(s) the authors are trying to convey. Similar comments can be made regarding figures 2, 3 and 4. The authors need to redo the figures and make them relevant to the discussion. Figure legends also need to be expanded.

2. The topics chosen to discuss do not seem to relate and it is unclear why the authors have discussion regarding changes in NO during oral contraception, Covid-19 and surgery (3 unrelated topics). This reviewer understands the discussion regarding HbNO but it remains unclear if these are the only conditions in which HbNO is measured.

3. The authors appear to provide anecdotal evidence regarding HbNO and vascular function but don't provide any studies that have attempted to related altered levels of HbNO to vascular function. Are there studies that have measured vascular function and relate this to altered levels of HbNO?

Author Response

Response 1: We have changed the quality of the graphics so that they are more informative and readable. Figure legends were already present but we have implemented them in such a way that they become more educational. We hope that these improvements will meet your expectations.

Response 2: We have tried to illustrate the scientific approach from the discovery of HbNO as an indicator of the NO-dependent endothelial function to various conditions known to present alterations of such endothelial function, e.g. under oral contraceptive pills or during COVID-19. Of course, surgery per se is not systematically a cause of the endothelial dysfunction. Nevertheless, a study is currently ongoing to explore the added value of HbNO as a biomarker to assess cardiovascular risk in patients undergoing a surgical operation with general anesthesia that, by itself, entails a cardiovascular stress (A Monocentric Study Assessing the Efficacy of the Nitrosylated Hemoglobin as Biomarker for Detecting the Development of a Cardiovascular Complication During or After Surgery, the PICA study). 

Response 3: The interest of this manuscript is precisely to demonstrate the existence of this original technique for evaluating the NO-dependent endothelial function, which is currently very difficult to estimate in clinical routine in patients. HbNO has indeed been correlated to vascular function (measured with the ENDO-PAT digital microplethysmography) in a previous pilot study in normal individuals (Lobysheva et al. PLoS One. 2013 Oct 10;8(10):e76457) and we recently published a correlation of HbNO with endothelial dysfunction in COVID-19 patients (Montiel et al. EBioMedicine. 2022 Mar;77:103893), but there are currently no published studies with large cohort of patients exploring the independent correlation of HbNO levels with vascular dysfunction. The ongoing PICA study will help clinicians to examine the added value of HbNO to refine cardiovascular risk independently from traditional risk factors.

Reviewer 3 Report

This work presents a comprehensive overview of the nitric oxide function and its assessment with regards to the endothelial cells. The work is well organized and it can attract the attention f the researchers from different fields. Some minor amendments should be taken into account before further steps:

1) The Figure 2 should contain the source and normally as this is the Fig from some paper it should have the authorization for the reprint

2) The quality of Fig2 seemed to be quite low, can you improve the resolution?

3) Please provide the graphical abstract to better attract the audience.

Author Response

Response: We thank the reviewer for his/her favorable opinion on the manuscript. We have changed the manuscript in accordance with his/her comments.

We have amended Figure 2, now of superior quality compared to the previous version; as it is now an original drawing made by us, its publication does not require any authorization for reprint.

We also added a graphical abstract, as requested.

Round 2

Reviewer 2 Report

No further comments.